# Predicting student satisfaction of emergency remote learning in higher education during COVID-19 using machine learning techniques

**Indy Man Kit Ho** **\***, **Kai Yuen Cheong, Anthony Weldon**

Technological and Higher Education Institute of Hong Kong (THEi), Chai Wan, Hong Kong

\* indyho@vtc.edu.hk

**Data Availability Statement:** Data is available upon request due to restrictions imposed by the Technological and Higher Education Institute of Hong Kong. For data access interested researchers

## Abstract

Despite the wide adoption of emergency remote learning (ERL) in higher education during the COVID-19 pandemic, there is insufficient understanding of influencing factors predicting student satisfaction for this novel learning environment in crisis. The present study investigated important predictors in determining the satisfaction of undergraduate students (N = 425) from multiple departments in using ERL at a self-funded university in Hong Kong while Moodle and Microsoft Team are the key learning tools. By comparing the predictive accuracy between multiple regression and machine learning models before and after the use of random forest recursive feature elimination, all multiple regression, and machine learning models showed improved accuracy while the most accurate model was the elastic net regression with 65.2% explained variance. The results show only neutral (4.11 on a 7-point Likert scale) regarding the overall satisfaction score on ERL. Even majority of students are competent in technology and have no obvious issue in accessing learning devices or Wi-Fi, face-to-face learning is more preferable compared to ERL and this is found to be the most important predictor. Besides, the level of efforts made by instructors, the agreement on the appropriateness of the adjusted assessment methods, and the perception of online learning being well delivered are shown to be highly important in determining the satisfaction scores. The results suggest that the need of reviewing the quality and quantity of modified assessment accommodated for ERL and structured class delivery with the suitable amount of interactive learning according to the learning culture and program nature.

## 1 Introduction

To date, the COVID-19 (2019–2020) outbreak has had widespread repercussions to the lifestyle and working mode for most people. Countries and cities continue to be locked down to promote social distancing and prevent excessive gatherings, controlling the further spread of COVID-19 [1]. Due to the unpredictable length of this pandemic, most education institutes have adopted emergency remote learning (ERL) via online learning platforms, to replace most, if not all, face-to-face theoretical and practical lessons [2–5]. Even programs traditionally taught in person have rapidly shifted to online learning, to fulfill education obligations and

may contact: Dr. Will Ma, Project Manager
willma@thei.edu.hk.

**Funding:** The authors received no specific funding for this work.

**Competing interests:** The authors have declared that no competing interests exist.

**Abbreviations:** ERL, Emergency remote learning; LMS, Learning management system; KNN, K-nearest neighbor; SVR, Support vector regression; RF, Random forest; GBM, Gradient boosting method; MLP, Multilayer perceptron; ENet, Elastic net; LASSO, Least absolute shrinkage and selection operator; RFE, Recursive feature elimination; MAE, Mean absolute error; RMSE, Root mean square error; RF-RFE, Random Forest Recursive Feature Elimination; M, Mean; SD, Standard deviation; $R^{2adj}$, Adjusted $R^2$.

avoid delaying students from graduating, pursuing further education, and obtaining employment.

Despite the emergent use of ERL, online learning is not something new. For example, it was reported that over six million (>30%) of students in the United States of America enrolled in at least one online course [6]. With the evolution of technology, various synchronous and asynchronous learning methods and blended learning have been developed and promoted, with the aid of web-based platforms and learning management systems (e.g., CANVAS, Moodle, Blackboard). Recently, with the vast enhancement in the internet and network technology, the integration of video conferencing and screen sharing tools (e.g., Microsoft Teams, Zoom, Google Meet) as a subset of online learning has been highly adopted and proposed to partially or entirely substitute face-to-face learning. This has enabled educators to simultaneously interact and monitor the learning progress of multiple students, whereas such enhancements have also potentially closed the gap in terms of learning quality, learning outcomes, and efficacy between face-to-face and online classes [7].

To investigate the success of online learning in higher education, numerous studies on student satisfaction have recently been conducted in different countries [2, 5, 8–16]. In the past decades, three main groups of factors including student (e.g., prior experience or knowledge, and self-efficacy), teacher (e.g., competencies in information and technology, information quality and feedback, and course structure), and technology acceptance and support (e.g., ease of use or access) related aspects have been addressed [11]. However, it is worth noting the substantial difference between online learning and ERL. Typically online learning is regarded as well-planned from the beginning and designed with a lengthy process. In contrast, ERL requires a hurried and temporary shift in instructional delivery due to crisis circumstances, leading to the complete closure of campuses [17–19]. Apart from the COVID-19 pandemic, ERL has also been adopted in regions suffering from conflict, violence, and war [18]. This use of fully remote online teaching and learning to continue the education that had been enforced, led to issues with lesson delivery and planned pedagogical methods. Therefore, unlike traditional online learning, the objective of ERL is to provide quick, temporary, and reliable access to teaching support. Limited studies have addressed student satisfaction during ERL, due to the recency of the pandemic [2, 5, 9, 10, 14, 20]. Therefore, the objective of this study is to examine higher education students' satisfaction during ERL.

Educational research regarding E-learning has shown considerable diversity in the statistical and research methods used [15]. The most simple, straightforward, and easy to interpret modeling methods may be multiple regression, which has been widely adopted in previous literature examining important factors influencing student satisfaction [21–30]. More recently the use of artificial intelligence-related data mining techniques, such as machine learning, for predicting students' performance in higher education has been extensively used [31]. Therefore, the secondary objective of this study is to compare the performance of machine learning and traditional multiple regression models. Meanwhile, this study will combine the use of machine learning algorithms and multiple regression to provide additional insights into the application of novel artificial intelligence techniques for future similar studies.

To reach a meaningful conclusion, the remainder of the paper is organized as follows: a review of existing literature; the research questions are presented with theoretical supports. Thereafter, the research methods including the rationales and benefits of using machine learning techniques, feature selection processes, a comparison of models, and sampling methods and procedures used to prepare for further analyses are described. Finally, the empirical results are demonstrated and discussed including the practical implications and limitations of this study.

## 2 Literature review

Numerous studies regarding online learning across higher education have been conducted, that have enhanced both the understanding and practical implications of adopting different modes of online learning, such as blended, asynchronous, and synchronous learning [15]. To determine the success of e-learning in higher education student satisfaction is an important indicator to determine performance [2, 5, 8–16]. Duque (2013) proposed a framework for evaluating higher education performance with students' satisfaction, perceived learning outcomes, and dropout intentions, and found that dropout intentions were strongly and negatively associated with student satisfaction [32]. Meanwhile, Kuo et al. (2014) highlighted the close relationship between student satisfaction and motivation, dropout rates, success, and learning commitment [33]. Furthermore, Pham et al. (2019) have shown a positive relationship between student satisfaction and loyalty in Vietnamese adults and higher education [13]. According to the E-learning systems success (EESS) model, it has been proposed that student satisfaction is a key component to determine E-learning success [8]. Therefore, through comprehensively understanding the underlying factors influencing student satisfaction, will enable the improvement of online teaching and learning design and execution [16].

### 2.1 Current theories for satisfaction in E-learning

Multiple factors have been proposed that identify and influence students' satisfaction regarding E-learning [8]. An early E-learning research model developed by DeLone and McLean (2003) was primarily based on the quality of information, systems, and services that determined user satisfaction [34]. This model has been used to compare E-learning success between male and female students in Malaysian universities during the COVID-19 pandemic [14]. Another significant approach for developing a theoretical framework in the research of E-learning is the user satisfaction approach [8]. A recent study conducted by Yawson and Yamoah (2020) adopted this approach using a 7-point Likert-scale, to measure the satisfaction of E-learning in higher education of developing countries (i.e., Ghana) [16]. Question items in their study included domains of the course design, delivery, interaction, and delivery environment. However, this study did not focus on ERL although the study period overlapped with the pandemic. Apart from the aforementioned models, other technology acceptance and E-learning quality models have been developed with an emphasis on usefulness and ease of use [8, 35]. Due to the unique characteristics, strength, and limitations in each research model, Al-Fraihat et al. (2020) has further formulated a multidimensional conceptual model for evaluating the EESS model more holistically [8].

Interestingly, a recent study by Shim and Lee (2020) developed a semi-structured questionnaire without adopting the aforementioned models to conduct a thematic analysis to investigate the colleges' experience of ERL during the COVID-19 pandemic in South Korea [5]. Similarly, Alqurshi (2020) used a tailor-made questionnaire to measure student's satisfaction using 5-point Likert-scale questions focusing on virtual classrooms, completion of course learning outcomes, and alternative assessments in different institutions in Saudi Arabia [10]. These previously mentioned theoretical models were built to evaluate pre-planned E-learning while the deployment of ERL during the COVID-19 pandemic was abrupt, direct use of E-learning research models may not suitably reflect the underlying factors affecting the success and satisfaction of ERL. Therefore, recently a tailor-made survey kit was developed by EDU-CAUSE to allow institutions to rapidly adopt to gather feedback from higher education stakeholders [36]. Therefore, the subsequent literature review has been primarily based on the items and constructs proposed in the EDUCAUSE survey kit, while taking reference from the components of the multidimensional EESS model.

## 2.2 Readiness and accessibility

The first part of the EDUCAUSE survey kit (2020) focuses on technological issues and challenges during the transition to remote learning [36]. Questions included the level of discomfort and familiarity of instructors and students while using technological applications, the adequacy of digital replacements for face-to-face collaboration tools (e.g., whiteboards), and accessibility to a reliable internet connection, communication software, and specialized software and tools. According to Al-Fraihat et al., (2020), the direct association between system quality and student satisfaction was assumed in the original model of Delone and Mclean (2003) [8, 34]. Similarly, other literature also suggests that improved system quality positively influences student satisfaction when E-learning [8, 37]. In the EESS model, the technical system quality has several subset items including ease of use and learning, user requirements, and the systems features, availability, reliability, fulfillment, security, and personalization. Whereas, Al-Fraihat et al. (2020) highlighted different obstacles when adopting E-learning in developing and developed countries [8]. For example, resources, accessibility, and infrastructure are more important for developing countries while information quality and usefulness of the system are more important in developed regions. However, low-income families may also exist in developed countries, and students from relatively poor living environments may face similar problems as those living in the developing countries, although the technological infrastructure of higher education institutes is better developed.

Also, self-efficacy, defined as the individuals' belief in their own ability to perform a certain task, challenge or successfully engage with educational technology [38, 39], showed to be interconnected with student satisfaction levels [40]. Recently, Prifti (2020) identified that the learning management system self-efficacy positively influenced student satisfaction in blended learning in Tirana of Albania while both platform content and accessibility were important constructs affecting the self-efficacy level [41]. Similarly, Geng et al. (2019) found technology readiness positively influenced learning motivation during blended learning in higher education [42]. Interestingly, Alqurashi (2018) reported conflicting findings regarding the impact of students' self-efficacy for using technology on student satisfaction, as more recent studies suggest university students have become more competent and confident in using technology when conducting online learning [43]. However, recently Rizun et al. (2020) confirmed that self-efficacy levels did affect students' acceptance in terms of perceived ease of use and usefulness when conducting ERL in Poland during the COVID-19 pandemic. Since the circumstances in well-planned and designed E-learning is different from ERL, it is important to assess important constructs such as accessibility and students' readiness, including their self-efficacy to determine ERL success [44].

## 2.3 Instructor, assessment, and learning

Another focus in the EDUCAUSE survey kit is learning and education-related issues. Focused questions include the personal preference for face-to-face learning, assessment requirements, students' attention to remote classes and activities, the availability and responsiveness of instructors, and if the original lessons were well translated to a remote format. Alqurashi (2018) showed the importance of quality learner-instructor interaction as two-way communication between the instructor and students [43]. Besides, his study used a multiple regression, which shown learner-content interaction was the most important predictor of student satisfaction, which further supports the findings from Kuo et al. (2014) study [33]. By providing user-friendly and accessible course materials, assists in the motivation of students' learning and understanding, in turn leading to increased student satisfaction. Meanwhile, the authors recommended students should pay more attention to the feedback and responses from the course

instructors, such as asking and answering questions, receiving feedback, and performing online discussions. Recently, Muzammil et al. (2020) demonstrated similar findings in Indonesian higher education using a structural equation model [12]. They showed that student-tutor interaction significantly contributed to the level of student engagement, whereas student satisfaction levels were greatly dictated by their engagement level. This was further demonstrated by Pham et al. (2019), who shown the instructor's ability to deliver quality E-learning provisions, affected Vietnamese college students' satisfaction and loyalty [13]. In their study, data regarding the perceived E-learning instructor quality from a students' perspective were gathered via several questions focusing on instructors knowledge, responsiveness, consistency in delivering good lectures, organization, class preparation, encouragement for interactive participation, and if the instructors have the students' best long-term interests in mind. However, recently in a review by Carpenter et al. (2020) raised the issue of students' "illusional learning", where well-polished lectures delivered by enthusiastic and engaging instructors can inflate students' subjective impressions and judgments of learning [45]. Since the evaluation of teaching effectiveness and quality of teachers from the students' point of view may have a strong bias, when designing a questionnaire concerning the instructor and E-learning for students, the focus should be placed on the familiarity in E-learning technology, responsiveness, and availability rather than teaching quality, performance and usefulness.

Based on the EESS model, the diversity in assessment materials significantly determines the educational system quality which contributes to the prediction of perceived satisfaction [8, 37]. Placing importance on assessments for predicting student satisfaction during E-learning was further supported by Hew et al. (2019) [26]. For example, using machine learning and hierarchical linear models, assessments were confirmed as a significant and important sentiment for predicting student satisfaction for MOOC (Massive Open Online Courses). Recently, Rodriguez et al. (2019) used multiple linear regression to determine assessment procedures and appropriate level of assessment demand as important predictors for student satisfaction levels in multiple universities from Andalusia, Spain [30]. When assessment-related aspects are considered while conducting ERL during a crisis, Shim and Lee (2020) identified comments regarding dissatisfaction with assessments, such as increasing the burden of final exams after the deletion of mid-term assessments, the vagueness of test evaluations, and increased quantities of assignments during COVID-19 [5]. As certain practical or tutorial classes might be moved to remote learning format or substituted by other learning activities, the change in assessment methods to accommodate such temporary shifts to ERL was necessary to match the actual learning quantity and quality of students. Therefore, the evaluation of the clarity and appropriateness of accommodated assessments seems to be essential in the prediction of students' satisfaction for ERL during a pandemic.

## 2.4 Self-concerned

Referring to the EESS model, learners' anxiety as part of the learner quality somewhat contributes to perceived satisfaction [8]. Bolliger and Halupa (2012), define anxiety as 'the conscious fearful emotional state' and further proposed the close relations between computer, internet, and online course anxiety [46]. In their study, a significant but negative association between student anxiety and satisfaction was detected, given several anxiety-related aspects such as performance insecurity, hesitation, and nervousness were proposed to closely link with student satisfaction. However, as Alqurashi (2018) emphasized the high computer and online learning competency of students nowadays, may limit the findings from prior studies addressing the effects of computer and internet related anxiety on students perceived satisfaction [43]. Therefore, suggested questions in the self-concern section of the EDUCAUSE survey kit include

other aspects potentially related to ERL, such as the worry about course performance or grade, the concern of lesser interaction with classmates and instructors, a potential delay of graduation or completion of the program, privacy, and food or housing security [36].

## 2.5 The application of multiple regression in predicting student satisfaction

Numerous researchers have used statistical methods to analyze satisfaction scores, perceived learning, interaction, self-efficacy, and other factors related to online learning [33, 43, 47–51]. For example, multiple linear regressions have been used to produce several predictive models for examining and comparing the interaction and amount of variance explained for different predictors on student satisfaction [43]. Multiple linear regression contains more than one independent variable ($X_1,\ldots,X_p$). It can be regarded as the expansion of a simple linear regression studying straight line mathematics with $Y = \beta_0 + \beta_1 X$ where $\beta_0$ is the intercept and $\beta_1$ is the slope. This statistical method has been widely used because of its simple algorithm and mathematical calculation [43, 52, 53]. Previous studies have shown its strong predictive power in applications but the estimated regression coefficients can be greatly affected if high correlations between predictors exist as the multicollinearity issue [54]. Apart from the simple linear regression, a hierarchical linear model was commonly used to deal with more complicated data with nested nature [26]. Meanwhile, stepwise multiple regression including the combination of the forward and backward selection techniques was widely adopted for high efficiency using the minimum number of important predictors to build a successful prediction model. However, numerous studies have pointed out the potential flaws using stepwise regression such as multicollinearity, overfitting, and the selection of nuisance variables rather than useful variables [55, 56]. Since only numerical variables are allowed for building predictive models in multiple linear regressions, categorical predictors including nominal and ordinal variables must be converted to binary code using dummy variables before modeling.

## 2.6 The use of machine learning

Opposed to multiple linear regressions, other machine learning methods under the umbrella of artificial intelligence are increasingly used for predictive purposes [52, 57–63]. The advantage of machine learning is the ability to use both categorical and numerical predictors to generate models through assessing linear and non-linear relationships between variables, and the importance of each predictor. Common machine learning algorithms for predicting numerical outcomes using regressors have been widely studied and adopted for applications of different contexts such as K-nearest neighbor (KNN) [57], support vector regression (SVR) [58, 61], an ensemble of decision trees with random forest (RF) [60], gradient boosting method (GBM) [62, 63], multilayer perceptron regression (MLPR) simulating the structure and operation of human neural network architecture [52], and elastic net (ENet) [64].

The KNN is a nonparametric method used to provide a query point for making predictions. Through computing the Euclidean distance between that point and all points in the training data set, the closest K training data points are picked. While the prediction is achieved by averaging the target output values for K points [57]. It is a simple machine learning method and easy to tune for optimization.

The SVR was developed as a supervised machine learning technique for handling classification problems. The SVR was later extended from the original support vector machine algorithm for solving multivariate regression problems [57, 61]. By constructing a set of hyperplanes in high-dimensional space, SVR makes the non-linear separable problem to be linearly separable [57, 61]. Therefore, SVR is a good option for solving problems with high

dimensional data with a lesser risk of overfitting, though it is sensitive to outliers and very time-consuming in training with large datasets.

The RF is a non-parametric method using an ensemble of decision trees with the voting of the most popular class while the results from trees are aggregated as the final output. In training the RF model, a multitude of decision trees is constructed using a collection of random variables [59, 60]. Random forest is broadly applicable to different populations due to being fast and efficient in generating predictions, with only a few parameters required to tune for model optimization. Moreover, it can be used for high-dimensional problems and provide feature importance for further analysis.

Unlike other tree-based machine learning techniques using level-wise learning to grow the tree vertically, LightGBM is an improved form of a gradient boosting algorithm. It uses a leaf-wise tree-based approach for enhancing the scalability and efficiency, with the lesser computational time required, and without sacrificing the model accuracy. Recent studies have shown excellent predictive performance in different data [62, 63].

The MLPR is a form of feedforward artificial neural network, simulating the structure and asynchronous activity of the human nervous system. With the input, hidden and output layers of nodes, neurons can perform nonlinear activation functions and distinguish non-linear data for supervised machine learning models [52, 57].

The ENet method was initially developed to simulate an elastic fishing net to retain "all the big fish", through automatic selection of predictors and continuous shrinkage. While the selection of a group of correlated variables is allowed, it provides both features and benefits of "ridge" and "least absolute shrinkage and selection operator (LASSO) regression". This was regarded as an improved form of multiple linear regression using ordinary least squares [65]. Recent studies demonstrated superior performance in using ENet over other regression methods in handling multicollinearity of predictors for numerical predictions [66, 67].

There is no best machine learning or statistical method for prediction accuracy, given the different structure and nature of datasets, including the number of variables, dimensionality, and cardinality of predictors, that can substantially influence the accuracy of each algorithm [52, 57, 60–63, 67]. Although previous studies have shown machine learning algorithms to outperform multiple linear regressions, especially in handling complicated models or datasets with high complexity, most machine learning methods are black box in nature and uninterpretable [68–70]. Consequently, the trade-off between prediction accuracy and capability in model explanation has become controversial for making decisions in using simple and transparent models like multiple linear regression or potentially more accurate but complicated black-box machine learning models. Recently Abu Saa et al. (2019) have highlighted the frequent use of machine learning techniques for educational data mining including Decision Trees, Naïve Bayes, artificial neural networks, support vector machine, and logistic regression [31]. Therefore, the use of machine learning algorithms in solving educational research problems such as student satisfaction can be a future exploratory direction.

## 2.7 Feature selection before building predictive models

To successfully build predictive models, feature selection is a critical and frequently used technique in both the field of statistics and machine learning to choose a subset of attributes from original features. This process attempts to reduce the high-dimensional feature space through the removal of redundant and irrelevant predictors and only select highly relevant features to enhance model performance [56, 71]. In addition to the automated selection process in stepwise regression, recursive feature elimination (RFE) is another commonly used feature selection method. Through repeatedly eliminating features in the lowest rank regarding the

relevancy and comparing the corresponding model accuracy after each RFE iteration, the subset of features/predictors is finalized for formulating the optimal model. In this regard, previous studies have shown the beneficial effect of wide-ranging RFE approaches in enhancing the prediction accuracy when building classification or regression predictive models after noise variables removed [71–74].

## 3 Research questions

The primary purpose of this study is to explore student satisfaction while conducting ERL during a pandemic and to identify relevant important predictors. Having taken references from the highly-specific EDUCAUSE survey kit for ERL during a crisis and the EESS model proposed by Al-Fraihat et al. (2020), six constructs including readiness, accessibility, instructor-related factors, assessment-related factors, learning-related factors, and self-concern were formulated for determining student satisfaction [8]. To minimize flaws and bias due to inappropriate selection of regression models, another purpose of the study is to compare the performance between machine learning and traditional multiple regression models with or without feature selection pre-processing. Therefore, research questions are presented as:

RQ1 –Does the feature selection method RFE enhance the prediction accuracy for selected statistical and machine learning models?

RQ2 –When comparing the multiple regression and stepwise regression with selected machine learning models (KNN, SVM, RF, LightGBM, MLPR, and ENet), which predictive model performs best?

RQ3 –To what extent do the predictors of six constructs (readiness, accessibility, instructor related factors, assessment-related factors, learning-related factors, and self-concern) predict student satisfaction using ERL during the COVID-19 pandemic?

## 4 Material and methods

This study adopted a quantitative research design to predict student satisfaction during online learning, throughout the COVID-19 pandemic. It was approved by the Human Research Ethics Committee of the Technological and Higher Education Institute of Hong Kong. The subjects of this study were undergraduate students at a self-funded Hong Kong university and selected by a purposeful sampling technique. Due to the different nature between well-planned online learning and ERL, a structured questionnaire was designed and modified from the questionnaire template provided by "EDUCAUSE DIY Survey Kit: Remote Work and Learning Experiences" [36]. As previous studies have highlighted the challenges in designing an appropriate survey for assessing student satisfaction such as appropriately controlling the number of predictors and dependent variables [8, 16], and the selection of relevant question items fit for the context of the experiment and environment, the survey must be carefully designed to accommodate all these challenges. Therefore, before distributing the questionnaire to students, a working group composed of teaching faculties and E-learning experts was formed to finalize the question items relevant for the predictive purpose. To facilitate a higher response rate and reduce the nonresponse bias, an anonymous survey containing only 27 items and additional questions for gathering demographic data were used so that students could complete the survey in about 10 to 15 minutes. In addition, the focus of studying student satisfaction on ERL and the purpose of the survey for future evaluation and enhancement on teaching and learning were clearly explained to motivate students in responding to the survey proactively. The

survey was distributed to a total of 3219 currently registered students in the institute through the learning management system Moodle, and was available for six-weeks between the 29 of April and 10 of June 2020, in semester two. In total 425 students (13.2%) responded to the online survey. The descriptive analysis of participants is presented in Table 1.

The survey used to assess students' satisfaction while conducting ERL during the COVID-19 pandemic included twenty-seven items from six constructs, including 1) readiness; 2); accessibility; 3) instructor related; 4) assessment-related; 5) learning-related and; 6) self-concern, and the dependent variable satisfaction of online learning. Participants rated their level of agreement on all items using a 7-point Likert-type scale, where 1 indicated strongly disagree, 2 disagree, 3 somewhat disagree, 4 neutral, 5 somewhat agree, 6 agree and 7 strongly agree. Table 2 shows the constructs and question items. Demographic data were also collected via questions regarding gender, mode of study, year of study, belonging faculty, self-rated competence on digital knowledge, the access of reliable Wi-Fi, and the type of device for conducting ERL.

## 4.1 Parameters tuning and modelling

All data were pre-processed with Microsoft Excel. After removing two rows of data with missing values, the final dataset was composed of 423 rows with 44 columns, including 27 question items, gender, year of study, faculty belonged to, and responses regarding the accessibility of technological resources and self-efficacy of using digital technology. The structure of the dataset is shown in Table 3. One-hot encoding was applied to convert features containing multiple categorical values to a binary numerical format to allow machine learning modeling [75]. After data pre-processing, conventional and stepwise multiple linear regression was developed with R in Rstudio (version 1.2.5001), while machine learning models including KNN, SVR, MLPR, LightGBM, RF, and ENet regression were built using Jupyter Notebook with the scikit-learn package in Python 3.

To avoid overfitting and inflated results when the model complexity was increased, the dataset was split into 10 independent sets of observations (K = 10) for all machine learning model development using K-fold cross-validation except statistical linear regression models [76, 77]. Eighty percent of data from the dataset was for training the model and 20% were unseen data for testing the performance. In performing machine learning models, MinMaxScaler was used to further normalize data for better model performance (i.e., data were scaled between 0 and 1) [78]. The performance of models was evaluated using mean absolute error (MAE), root means square error (RMSE), and coefficient of determination ($R^2$). The formulae

**Table 1. Descriptive analysis of participants completed the online survey.**

| Description | Frequency (%) |
|---|---|
| Gender | • Male: 207 (48.7%) |
|  | • Female: 218 (51.3%) |
| Mode of Study | • Full-Time: 413 (97.2%) |
|  | • Part-Time: 12 (2.8%) |
| Year of Study | • Year 1: 60 (14.1%) |
|  | • Year 2: 74 (17.4%) |
|  | • Year 3: 175 (41.2%) |
|  | • Year 4: 112 (26.4%) |
|  | • Other: 4 (0.9%) |

**Table 2. Survey items and constructs.**

| Constructs | Items | Description |
|---|---|---|
| Readiness | Q2 | I am comfortable or familiar with the required technologies or applications. |
| | Q3 | I am clear which technologies and applications I am required to use. |
| | Q4 | I have no difficulty accessing reliable communication software/tools (e.g., MS Teams, Zoom, Google Hangout). |
| Accessibility | Q5 | I can always access specialized software for my study (e.g., Adobe products, statistical packages). |
| | Q6 | I can always access library resources. |
| | Q7 | I can always find time to participate in synchronous classes (e.g., live-streaming lectures or video conferencing at a set time). |
| | Q9 | I can always find time for class meetings and schedules. |
| Instructor related | Q1 | In general, my instructors are comfortable or familiar with the required technologies or applications. |
| | Q13 | In general, my instructors are available or responsive. |
| | Q19 | The instructors in the program made efforts to enhance my learning. |
| | Q20 | During this transition period of the program, students' suggestions and comments were listened to and acted upon appropriately. |
| Assessment related | Q8 | I am clear about my course/assignment requirements |
| | Q16 | Methods of adjusted/modified assessment in this transition period are appropriate for evaluating my achievement of the intended learning outcomes |
| | Q17 | The criteria used for adjusted/modified assessment marking were clear to me. |
| | Q18 | The adjusted/modified assessment was effective in helping me learn. |
| Learning-related | Q10 | I prefer face-to-face learning. |
| | Q11 | I feel that my course lessons or activities have been well delivered in the online environment. |
| | Q12 | I can always focus on or pay attention to remote instructions or activities. |
| | Q14 | I am motivated / I desire to complete my coursework. |
| | Q15 | The online learning materials (Zoom/Team/video) facilitated my learning. |
| Self-concern | Q21 | I am concerned about my grades/performing well in class. |
| | Q22 | I am concerned about the changes to the grading structures (e.g., pass/fail). |
| | Q23 | I am concerned about possible delays in graduating/completing my program. |
| | Q24 | I am concerned about my instructors not using Moodle/Canvas. |
| | Q25 | I am concerned about my instructors using a tool that is not supported by the institution. |
| | Q26 | I am concerned about my instructors not recording online lessons delivered and making the videos accessible to students thereafter. |
| Outcome: Satisfaction | Q27 | Overall, I am satisfied with online learning in the last three months. |

for calculation of MAE, RMSE, and R$^2$ are as follow:

$$\text{RMSE} = \sqrt{\frac{1}{N}\sum\nolimits_{i=1}^{N}(O_i - P_i)^2} \tag{1}$$

$$\text{MAE} = \frac{1}{N}\sum\nolimits_{i=1}^{N}|(O_i - P_i)| \tag{2}$$

$$\text{R}^2 = 1 - \frac{\sum (O_i - P_i)^2}{\sum (O_i - P_m)^2} \tag{3}$$

**Table 3. Structure of dataset.**

| Predictors | Description |
|---|---|
| Gender | "1" = male; "0" = female |
| Year_1 | "1" for year 1 participants; "0" for others |
| Year_2 | "2" for year 2 participants; "0" for others |
| Year_3 | "3" for year 3 participants; "0" for others |
| Year_4 | "4" for year 4 participants; "0" for others |
| Year_5 | "5" for participants beyond year 4; "0" for others |
| FDE | "1" for Faculty of Design and Environment (FDE); "0" for others |
| FMH | "1" for Faculty of Management and Hospitality (FMH); "0" for others |
| FST | "1" for Faculty of Science and Technology (FST); "0" for others |
| Others_faculty | "1" for participants not included in FDE, FMH, FST; "0" for others |
| Wi-Fi | "1" for participants having reliable access to Wifi at home; "0" indicates no Wifi |
| Desk_laptop | "1" for participants having a reliable computer at home; "0" indicates no reliable computer use at home |
| Sole_use_computer | "1" for the sole use of computer; "0" indicates not |
| Mobile | "1" for using while "0" indicates no use of such device for online learning |
| Tablet | |
| Computer | |
| Others | |
| Digital_knowledge | Numerical value to self-rate the digital knowledge and skill from 1 (minimum) to 10 (maximum). |
| Q1_Instructor_familiar_tech_apps | Numerical responses using 7-point Likert-type scale (1 to 7) where 1 indicated strongly disagree, 2 for disagree, 3 for somewhat disagree, 4 for neutral, 5 for somewhat agree, 6 for disagree, and 7 for strongly agree for question 1 to 27 shown in Table 2 |
| Q2_Self_comfort_familiar_tech_apps | |
| Q3_Self_clear_techapp_selection | |
| Q4_Self_accessible_reliable_communication_software | |
| Q5_Self_access_specialized_software | |
| Q6_Self_access_library_resource | |
| Q7_Self_find_time_in_synchronous_classes | |
| Q8_Self_clear_course_assignment_requirement | |
| Q9_Self_find_time_for_class_meetings | |
| Q10_Self_prefer_face_to_face_learning | |
| Q11_Self_courses_activities_well_delivered_in_online | |
| Q12_Self_focus_attention_to_remote_instruction | |
| Q13_Instructor_available_responsive | |
| Q14_Self_motivated_or_desire_to_complete_coursework | |
| Q15_Online_materials_facilitate_learning | |
| Q16_Adjusted_assessment_appropriate_for_evaluate_outcomes | |
| Q17_Criteria_adjusted_assessment_marking_clear | |
| Q18_Adjusted_assessment_effective_help_learning | |
| Q18_Adjusted_assessment_effective_help_learning | |
| Q19_Instructors_made_effort_enhance_learning | |
| Q20_Students_suggestions_comments_listened_acted_appropriate | |
| Q21_Self_concern_grade_performing_well_inclass | |
| Q22_Self_concern_changes_to_grading_structure | |
| Q23_Self_concern_delay_graduation_programme_completion | |
| Q24_Self_concern_instructor_not_using_LMS_moodle | |
| Q25_Self_concern_instructor_using_tools_software_not_supported_institute | |
| Q26_Self_concern_no_online_lesson_recording_and_video_access | |
| Q27_Satisfaction_online_learning | |

Where N is the number of data points, $O_i$ and $P_i$ are the observed and predicted values respectively, and $P_m$ is the mean of $P_i$ values. The final MAE, RMSE, and $R^2$ values were calculated by averaging the corresponding values from 10 sets of cross-validation in training and testing sets for the corresponding accuracy. When the training accuracy of a model substantially outperformed the testing one, it was considered as overfitting and further reiteration was performed after re-tuning of hyper-parameters. Conversely, models were regarded as underfitting if the testing accuracy were higher than the training one. The model with the highest testing accuracy without obvious underfitting and overfitting was regarded as optimum for further analyses and comparisons.

The building of RF, MLPR, KNN, LightGBM, and ENet models were optimized through manual tuning of parameters as research suggests the higher efficiency and potentially better performance using a random manual search over a grid search technique [79]. Conversely, SVR was optimized through a grid search function due to the time-consuming computation and tuning processes for multiple parameters. All models were repeatedly reiterated until the highest testing accuracy in both MAE and RMSE yielded without substantial overfitting.

### 4.2 Random Forest Recursive Feature Elimination (RF-RFE)

In addition to interpreting performance for selected predictive models using all features shown in Table 3, feature selection with the RF-RFE technique was applied. This removed redundant and irrelevant information for decreasing the data dimensionality. To find the optimum subset of features, RF was used to train the model based on the training data [73]. The ranking of feature importance was acquired and sorted after RF modeling, and the two least important features were removed, while the subsequently updated features were re-trained with RF models again. By repeating the above procedures until no further feature subset remained, the model accuracy in MAE for each subset of the feature was obtained for comparison. The feature subset with the lowest MAE value was used for further machine learning modeling. If more than one subset of features yielded the best performance, the one with the least features was used.

## 5 Results

In total 425 students responded to the survey, with demographic information presented in Table 1). Gender distribution of respondents were fairly equal (Male: 207, 48.7%; female: 218, 51.3%). A total of 88 (20.7%), 127 (29.9%) and 202 (47.3%) respondents were from the Faculty of Design and Environment, the Faculty of Management and Hospitality, and the Faculty of Science and Technology respectively. Most students had reliable access to Wi-Fi (90.8%) and computers (91.5%) at home. Whereas, 66.6% primarily used a desktop or laptop to conduct online learning, and 22.4% a mobile phone. Students' average self-rated digital knowledge was 6.36 out of 10, where 1 and 10 indicate the minimum and maximum ratings respectively.

Tables 4 and 5 show the descriptive results including the mean (M) scores, standard deviations (SD), skewness, and kurtosis for the 27 question items, as well as the internal reliability of the survey constructs. The normality was assumed as both skewness and kurtosis were within +2.0 and -2.0 [80]. The highest mean score of the five predictors was Q21 (M = 5.66, SD = 1.38), Q22 (M = 5.54, SD = 1.35), Q23 (M = 5.47, SD = 1.49), Q10 (M = 5.12, SD = 1.67) and Q26 (M = 5.02, 1.58), with the lowest being Q11 (M = 3.91, SD = 1.62). The result of the dependent variable (Q27) regarding satisfaction scores of online learning in the last three months was 4.11 (SD = 1.67). The survey scale is deemed reliable as the Cronbach's alpha values of all constructs are greater than 0.70 [81].

**Table 4. Descriptive statistics of the score on 27 question items.**

| Items | Description | Mean | SD | Skewness | Kurtosis |
|---|---|---|---|---|---|
| Q1 | In general, my instructors are comfortable or familiar with the required technologies or applications. | 4.36 | 1.442 | -0.52 | -0.33 |
| Q2 | I am comfortable or familiar with the required technologies or applications. | 4.69 | 1.418 | -0.59 | -0.15 |
| Q3 | I am clear about which technologies and applications I am required to use. | 4.95 | 1.350 | -0.83 | 0.42 |
| Q4 | I have no difficulty accessing reliable communication software/tools (e.g., MS Teams, Zoom, Google Hangout). | 4.74 | 1.547 | -0.63 | -0.38 |
| Q5 | I can always access specialized software for my study (e.g., Adobe products, statistical packages). | 4.31 | 1.533 | -0.33 | -0.54 |
| Q6 | I can always access library resources. | 4.23 | 1.521 | -0.39 | -0.39 |
| Q7 | I can always find time to participate in synchronous classes (e.g., live-streaming lectures or video conferencing at a set time). | 4.45 | 1.613 | -0.41 | -0.71 |
| Q8 | I am clear about my course/assignment requirements. | 4.20 | 1.615 | -0.28 | -0.93 |
| Q9 | I can always find time for class meetings and schedules. | 4.49 | 1.547 | -0.43 | -0.57 |
| Q10 | I prefer face-to-face learning. | 5.12 | 1.665 | -0.76 | -0.04 |
| Q11 | I feel that my course lessons or activities have been well delivered in the online environment. | 3.91 | 1.622 | -0.24 | -0.86 |
| Q12 | I can always focus on or pay attention to remote instructions or activities. | 4.02 | 1.501 | -0.26 | -0.51 |
| Q13 | In general, my instructors are available or responsive. | 4.65 | 1.524 | -0.58 | -0.19 |
| Q14 | I am motivated / I desire to complete my coursework. | 4.45 | 1.581 | -0.41 | -0.50 |
| Q15 | The online learning materials (Zoom/Team/video) facilitated my learning. | 4.22 | 1.525 | -0.45 | -0.42 |
| Q16 | Methods of adjusted/modified assessment in this transition period are appropriate for evaluating my achievement of the intended learning outcomes. | 4.09 | 1.525 | -0.35 | -0.40 |
| Q17 | The criteria used for adjusted/modified assessment marking were clear to me. | 4.13 | 1.487 | -0.36 | -0.49 |
| Q18 | The adjusted/modified assessment was effective in helping me learn. | 4.12 | 1.481 | -0.41 | -0.53 |
| Q19 | The instructors in the program made efforts to enhance my learning. | 4.47 | 1.514 | -0.45 | -0.41 |
| Q20 | During this transition period of the program, students' suggestions and comments were listened to and acted upon in an appropriate way. | 4.35 | 1.412 | -0.38 | -0.08 |
| Q21 | I am concerned about my grades/performing well in class. | 5.66 | 1.380 | -1.30 | 1.75 |
| Q22 | I am concerned about the changes to the grading structures (e.g., pass/fail). | 5.54 | 1.346 | -1.09 | 1.31 |
| Q23 | I am concerned about possible delays in graduating/completing my program. | 5.47 | 1.489 | -1.14 | 1.12 |
| Q24 | I am concerned about my instructors not using Moodle/Canvas. | 4.63 | 1.583 | -0.40 | -0.33 |
| Q25 | I am concerned about my instructors using a tool that is not supported by the institution. | 4.48 | 1.531 | -0.39 | -0.19 |
| Q26 | I am concerned about my instructors not recording online lessons delivered and making the videos accessible to students thereafter. | 5.02 | 1.577 | -0.65 | 0.02 |
| Q27 | Overall, I am satisfied with the online learning in the last three months. | 4.11 | 1.665 | -0.32 | -0.77 |

All features including demographic data and results from 27 question items were pre-processed and Fig 1 presents the results of the RF-RFE. The highest accuracy was observed after 28 features were removed (MAE = 0.615 when 16 features were retained). The final 16 selected features are shown in Table 6.

To predict student satisfaction for ERL during COVID-19, conventional multiple linear regression, stepwise multiple linear regression, and multiple machine learning models

**Table 5. Internal reliability of survey constructs.**

| Constructs | Cronbrah's alpha (95% confidence limits) | Classification |
|---|---|---|
| Readiness | 0.83 (0.81, 0.86) | Good |
| Accessibility | 0.76 (0.73, 0.80) | Acceptable |
| Instructor related | 0.86 (0.83, 0.88) | Good |
| Assessment related | 0.89 (0.87, 0.91) | Good |
| Learning related (with Q10 score reversed) | 0.78 (0.75, 0.82) | Acceptable |
| Self-concerned | 0.81 (0.78, 0.84) | Good |

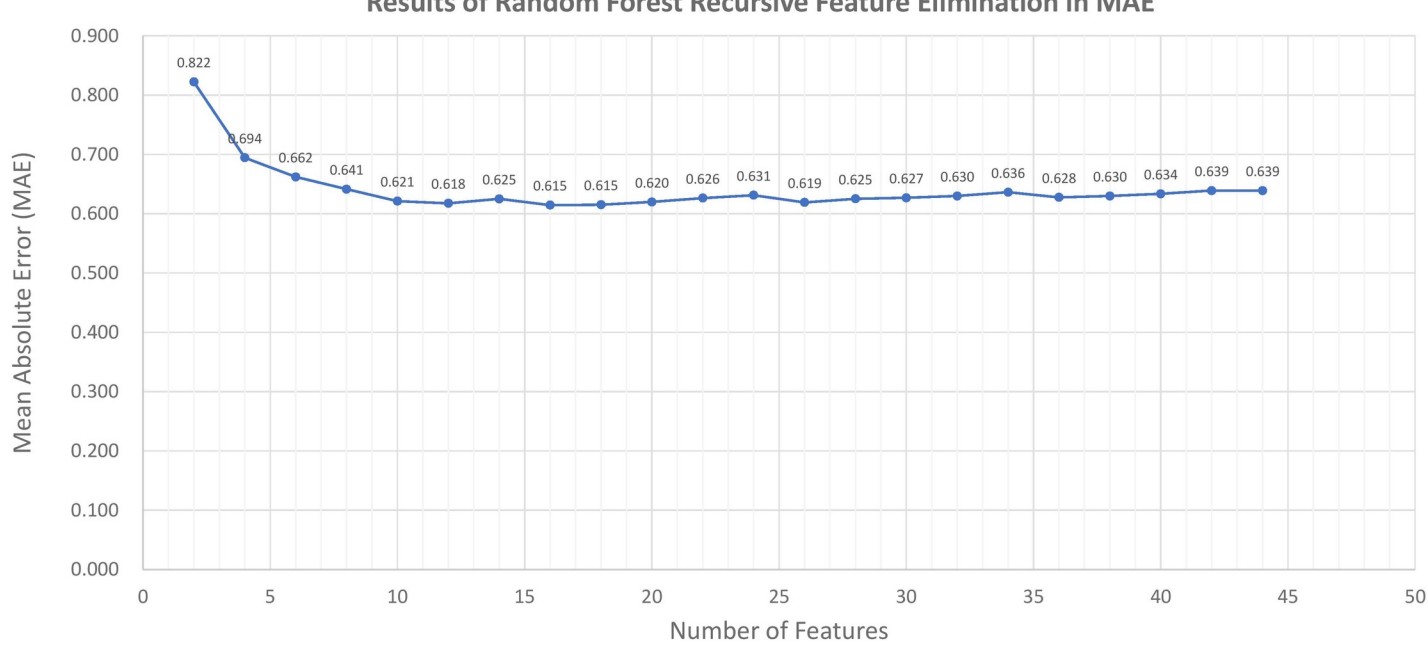

**Fig 1. Graphical presentation of results of recursive feature elimination using random forest algorithm.**

including KNN, SVR, MLPR, RF, LightGBM, and ENet were conducted. This provided a comparison of the accuracy of MAE and RMSE, and $R^2$ values before and after using the RF-RFE feature selection technique. Table 7 and Fig 2 show all selected models produced higher accuracy in both MAE and RMSE values after using the RF-RFE technique. For the MAE of testing data, the best performing models after RF-RFE were MLPR (0.729), ENet (0.744), conventional multiple linear regression (0.755), and LightGBM (0.755), while KNN performed poorest (MAE = 0.787). For the RMSE of testing data, the best performing models were ENet (0.968),

**Table 6. The remained features after recursive feature elimination with random forest.**

| Features retained | |
|---|---|
| Readiness | Q2_Self_comfort_familiar_tech_apps |
| | Q3_Self_clear_techapp_selection |
| Accessibility | Q9_Self_find_time_for_class_meetings |
| Instructor-related | Q1_Instructor_familiar_tech_apps |
| | Q13_Instructor_available_responsive |
| | Q19_Instructors_made_effort_enhance_learning |
| | Q20_Students_suggestions_comments_listened_acted_appropriate |
| Assessment-related | Q8_Self_clear_course_assignment_requirement |
| | Q16_Adjusted_assessment_appropriate_for_evaluate_outcomes |
| | Q17_Criteria_adjusted_assessment_marking_clear |
| | Q18_Adjusted_assessment_effective_help_learning |
| Learning-related | Q10_Self_prefer_face_to_face_learning |
| | Q11_Self_courses_activities_well_delivered_in_online |
| | Q12_Self_focus_attention_to_remote_instruction |
| | Q14_Self_motivated_or_desire_to_complete_coursework |
| | Q15_Online_materials_facilitate_learning |

**Table 7. Results of predictive models with and without recursive feature elimination.**

| | | MAE | | RMSE | | $R^2$ | |
|---|---|---|---|---|---|---|---|
| | | Training | Testing | Training | Testing | Training | Testing |
| No RF-RFE* | Multiple linear regression | 0.683 | 0.790 | 0.883 | 1.062 | 0.721 | 0.592 |
| | Stepwise multiple linear regression | 0.702 | 0.787 | 0.903 | 1.070 | 0.708 | 0.588 |
| | KNN | 0.812 | 0.961 | 1.017 | 1.202 | 0.628 | 0.467 |
| | SVR | 0.768 | 0.785 | 0.964 | 1.007 | 0.665 | 0.625 |
| | Multilayer Perceptron | 0.702 | 0.757 | 0.931 | 1.009 | 0.688 | 0.622 |
| | LightGBM | 0.634 | 0.759 | 0.816 | 0.997 | 0.760 | 0.631 |
| | RF | 0.721 | 0.834 | 0.905 | 1.056 | 0.705 | 0.589 |
| | ENet | 0.719 | 0.754 | 0.930 | 0.984 | 0.688 | 0.641 |
| RF-RFE | Multiple linear regression | 0.715 | 0.755 | 0.930 | 1.003 | 0.690 | 0.637 |
| | Stepwise multiple linear regression | 0.724 | 0.763 | 0.934 | 1.018 | 0.687 | 0.625 |
| | KNN | 0.645 | 0.787 | 0.858 | 1.036 | 0.735 | 0.601 |
| | SVR | 0.745 | 0.764 | 0.948 | 0.985 | 0.676 | 0.640 |
| | Multilayer Perceptron | 0.718 | 0.729 | 0.958 | 0.976 | 0.670 | 0.646 |
| | LightGBM | 0.641 | 0.755 | 0.827 | 0.993 | 0.753 | 0.634 |
| | RF | 0.615 | 0.784 | 0.785 | 1.022 | 0.778 | 0.614 |
| | ENet | 0.728 | 0.744 | 0.944 | 0.968 | 0.679 | 0.652 |

*RF-RFE: Random forest recursive feature elimination

MLPR (0.976), SVR (0.985), and LightGBM (0.993), while the conventional multiple linear regression (RMSE = 1.003) and stepwise multiple linear regression (RMSE = 1.018) only showed fair accuracy in this regard. When $R^2$ values were compared for the testing data, ENet

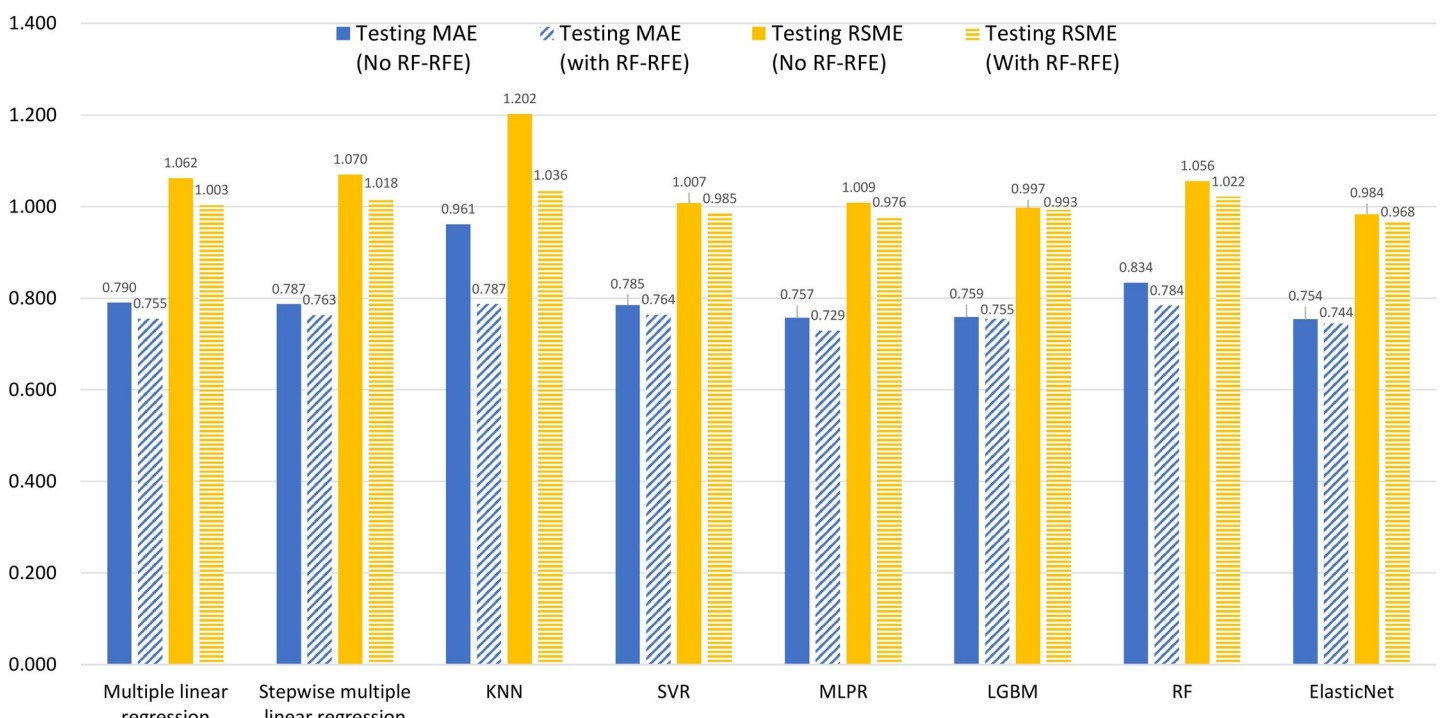

**Fig 2. Bar graph of Mean Absolute Error (MAE) and Root Mean Square Error (RMSE) of predictive models with and without using random forest recursive feature elimination.**

**Table 8. Standardized coefficient beta of the multiple linear regression model after recursive feature elimination using random forest.**

| Constructs | Items | Beta | t | Sig. |
|---|---|---|---|---|
|  | (Constant) | 0.955 | 3.379 | 0.000 |
| Instructor | Q1_Instructor_familiar_tech_apps | 0.084 | 1.560 | 0.120 |
| Readiness | Q2_Self_comfort_familiar_tech_apps | 0.085 | 1.502 | 0.134 |
| Readiness | Q3_Self_clear_techapp_selection | 0.018 | 0.308 | 0.758 |
| Assessment | Q8_Self_clear_course_assignment_requirement | 0.066 | 1.338 | 0.182 |
| Accessibility | Q9_Self_find_time_for_class_meetings | 0.065 | 1.410 | 0.160 |
| Learning | Q10_Self_prefer_face_to_face_learning | -0.274 | -8.124 | 0.000** |
| Learning | Q11_Self_courses_activities_well_delivered_in_online | 0.168 | 3.298 | 0.001** |
| Learning | Q12_Self_focus_attention_to_remote_instruction | -0.032 | -0.593 | 0.553 |
| Instructor | Q13_Instructor_available_responsive | 0.036 | 0.701 | 0.484 |
| Learning | Q14_Self_motivated_or_desire_to_complete_coursework | 0.068 | 1.464 | 0.144 |
| Learning | Q15_Online_materials_facilitate_learning | 0.042 | 0.706 | 0.481 |
| Assessment | Q16_Adjusted_assessment_appropriate_for_evaluate_outcomes | 0.161 | 2.744 | 0.006** |
| Assessment | Q17_Criteria_adjusted_assessment_marking_clear | 0.098 | 1.552 | 0.122 |
| Assessment | Q18_Adjusted_assessment_effective_help_learning | 0.052 | 0.761 | 0.447 |
| Instructor | Q19_Instructors_made_effort_enhance_learning | 0.133 | 2.252 | 0.025* |
| Instructor | Q20_Students_suggestions_comments_listened_acted_appropriate | 0.026 | 0.482 | 0.630 |

produced the best result (0.652) followed by the MLPR (0.646) and SVR (0.640), while the conventional and stepwise multiple linear regression yielded relatively lower values with 0.637, and 0.625 respectively.

Table 8 shows the multiple linear regression results after using RF-RFE. The observed overall model built with training data set demonstrated four predictors of interaction significantly predicted student satisfaction scores of online learning, $R^2 = 0.690$, Adjusted $R^2$ ($R^{2adj}$) = 0.675, F(16, 321) = 44.69, p < 0.00. This indicates that 69.0% of the variance could be explained through the model, meanwhile, the $R^2$ value dropped to 0.637 (7.68% decrement) when this built model was used for predicting unseen testing data. Among the 16 selected predictors, only four were shown to be significant in predicting student satisfaction including:

- Learning-related, Q10_Self_prefer_face_to_face_learning (β = -0.274, p < 0.00),

- Learning-related, Q11_Self_courses_activities_well_delivered_in_online (β = 0.168, p < 0.00),

- Assessment-related, Q16_Adjusted_assessment_appropriate_for_evaluate_outcomes (β = 0.161, p < 0.00) and;

- Instructor-related, Q19_Instructors_made_effort_enhance_learning (β = 0.133, p = 0.025).

The regression formula can be presented as follow:

$$0.955 + 0.133 \ (score \ of \ Q19) + 0.161 \ (score \ of \ Q16) + 0.168 \ (score \ of \ Q11) - 0.274 \ (score \ of \ Q10)$$

For the best performing machine learning algorithm after RF-RFE employed, ENet, the $R^2$ value of 0.679 for the training data indicates that this model accounted for 67.9% of the variance while the $R^2$ value for the testing data set was kept at 0.652 (3.98% decrement). Fig 3 shows the graphical ranking for feature importance of the model. The top four predictors with the largest weight coefficients were:

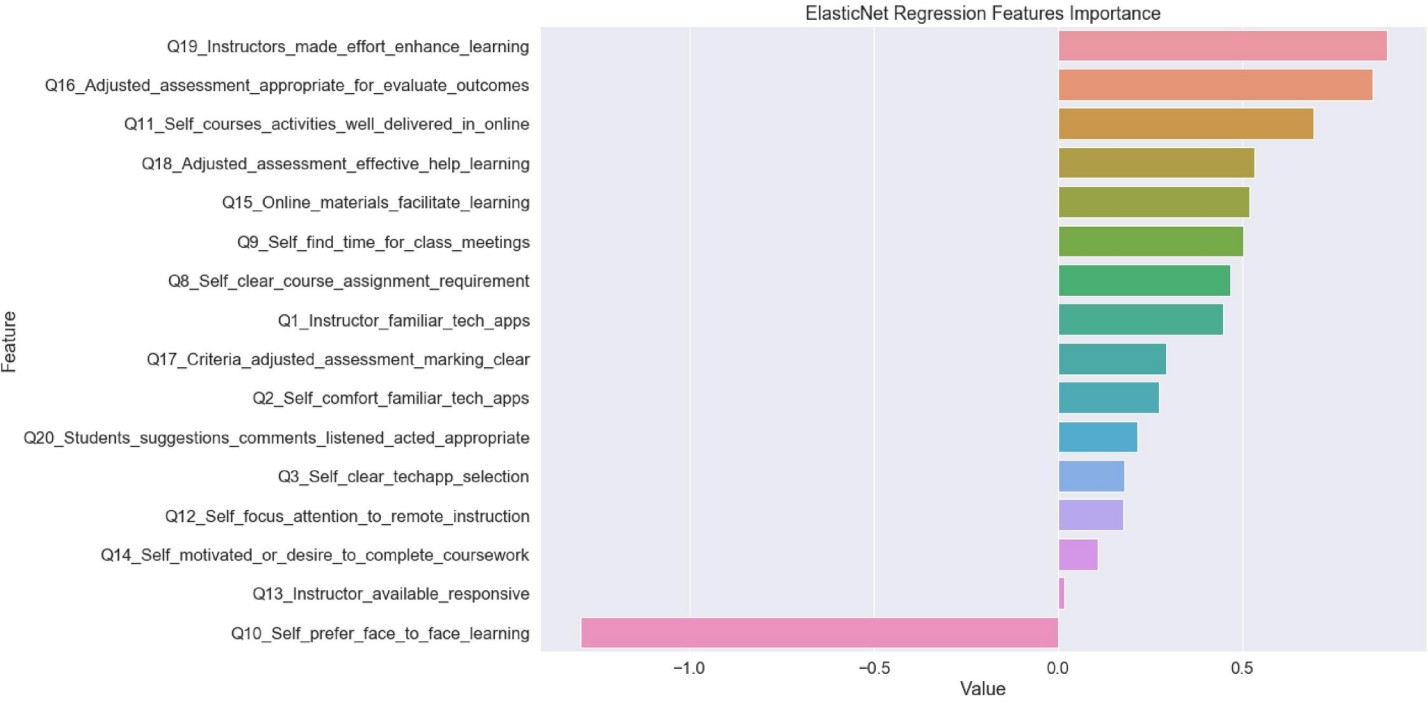

**Fig 3. Bar graph for feature importance of elastic net.**

- Learning related, Q10_Self_prefer_face_to_face_learning (-1.30),

- Instructor related, Q19_Instructors_made_effort_enhance_learning (0.890),

- Assessment related, Q16_Adjusted_assessment_appropriate_for_evaluate_outcomes (0.852) and;

- Learning related, Q11_Self_courses_activities_well_delivered_in_online (0.692).

## 6 Discussion

### 6.1 The value of feature selection with RF-RFE

When deciding the best statistical or machine learning model, it is believed that feature selection is an important procedure to remove irrelevant or redundant predictors that inflate the standard error of the estimated regression coefficients and decrease model performance. Without proper model optimization processes such as parameter tuning and feature selection, simple statistical models including linear regression are likely overfitted. This leads to poor prediction capability when the model is fit to the noise and the number of predictors was closed to the sample size [54]. The function and value of feature selection prior to modeling was evident in this study. Before using RF-RFE, both conventional (training MAE: 0.683; testing MAE: 0.790; training RMSE: 0.883; testing RMSE: 1.062) and stepwise multiple linear regression (training MAE: 0.702; testing MAE: 0.787; training RMSE: 0.903; testing RMSE: 1.070) showed a certain degree of overfitting as the MAE and RMSE values of the training set was superior to (lower MAE but higher RMSE values) that of the testing set. In this regard, an inflated value of explained variance was observed in both traditional statistical models (training and testing $R^2$: 0.721 and 0.592 for linear regression; 0.708 and 0.588 for stepwise

regression). However, after the RF-RFE was applied on both regression models, the accuracy difference between the training and testing tests was narrowed.

Furthermore, the stepwise multiple linear regression had no further improvement from the conventional multiple linear regression after using RF-RFE. It is apparent that RF-RFE had selected the most useful and relevant predictors, therefore, similar feature selection processes in stepwise regression did not provide additional benefits for model optimization. Consequently, using the $R^2$ value from the training set without a prior feature selection process and verification of accuracy from the testing set could lead to an overestimated model accuracy with the inflated values of explained variance. Apart from multiple linear regression, results showed after using RF-RFE, the accuracy in terms of MAE, RMSE, and $R^2$ of all selected machine learning models in predicting testing data set were improved. Similar to prior research the positive benefits of using RFE have been observed despite using different RFE techniques [71–74]. Therefore, our findings well respond to the RQ1 and support using RF-RFE feature selection as a pre-processing technique for similar studies.

## 6.2 The selection of predictive models

To the authors' knowledge, no previous study adopted machine learning algorithms in predicting student satisfaction for ERL or online learning, therefore, it was inconclusive if machine learning could provide higher prediction accuracy over traditional statistical linear regression in relevant domains. However, numerous previous studies showed the potential superior predictive performance using MLPR as an artificial neural network over the statistical multiple linear regression in wide-ranging applications such as food quality [82], climate prediction processes [52], behavior and deformation of dams [83], and epidemiological data [53]. Furthermore, Abu Saa et al. (2019) highlighted many emerging educational data mining studies using machine learning [31]. When comparing the MAE, RMSE, and $R^2$ values of the testing set, both ENet and MLPR outperformed conventional and stepwise multiple linear regression models. Since linear regression is only robust to a simple dataset with numerical predictors showing linear relationships, when categorical variables are included and model complexity increases, multiple linear regression may not best explain the complex relationship between predictors and the dependent variable especially when non-linear relationships exist [83]. Therefore, the results of this study provide empirical evidence in using machine learning algorithms to more accurately predict student satisfaction concerning ERL.

Nevertheless, deep learning techniques such as MLPR are regarded as "black box", which can limit further analysis on features' importance or the logic of the algorithm leading to prediction [84]. Therefore, with the good prediction accuracy of both MLPR and ENet observed in this study, ENet is considered as a better modeling choice. Since one-hot encoding was applied in our dataset, the model complexity increased substantially when dummy variables for categorical predictors with multiple values were produced. Given that model accuracy and stability can be negatively impacted in multiple linear regression for the inclusion of redundant variables highly correlated with each other [54], model tuning through regularization may help optimize the predictive accuracy. Elastic net is a regularized linear regression model providing different functions in ridge regression and LASSO for squared penalization and absolute value penalization respectively. By adjusting the hyper-parameter "l1_ratio", the alpha value between 0 and 1, the relative contribution of ridge regression and LASSO is selected. In our case, when setting the alpha value to 0.1 (higher contribution from ridge regression), the highest model performance was reflected through the accuracy of the testing set (MAE = 0.744; RMSE = 0.968), while both training and testing accuracy was similar, showing no obvious overfitting problem. On top of using RF-RFE, ENet provided additional benefits in

handling overfitting issues compared to conventional or stepwise multiple linear regression. Therefore, in response to RQ2, combining the RF-RFE technique and ENet for best features selection and predictive modeling is highly recommended for future studies regarding the prediction of student satisfaction scores for ERL with a comparable structure of the dataset.

## 6.3 Important predictors

In response to RQ3, since only four predictors were shown to be significant in predicting the dependent variable in linear regression while coincidentally, the top four important predictors (Q10, Q11, Q16, and Q19) derived from ENet regression also showed similar corresponding results, only these four items are discussed in the following parts.

Overall speaking, our study only showed "neutral" (4.11 out of 7) in terms of student satisfaction scores. Based on the results from ENet regression, "Q10 students preferring face-to-face learning" is the most impactful predictor while students scored on the high side for this question (5.12 out of 7, agree) which indicates the relatively low preference on ERL. From the technical issues point of view, over 90% of students had reliable Wi-Fi and the majority of them reflected relatively high self-efficacy on digital competence (6.36 out of 10), meanwhile infrastructure was fully supported with Moodle platform and video conferencing software Microsoft Team. Since students were already getting used to blended learning before the pandemic with training on the use of learning management system while the institute also provided tutorials and guidance for using Microsoft Team during the pandemic, neither the digital competence, accessibility to learning devices, and Wi-Fi related factors were shown to be significant in our predictive model. Therefore, the technical barrier or underdeveloped infrastructure was not likely the key factor contributing to the low satisfaction score on ERL. This supported the finding observed from another recent study conducted by Fatani (2020) that technical issues (audio/visual) were not the significant determinants of student satisfaction using ERL during the pandemic [20]. Both studies somewhat echoed and associated with the idea raised by Alqurashi et al. (2018) that computer and internet self-efficacies were not important in determining student satisfaction in online learning [43].

When the cultural differences in students' learning between western countries and China were considered, Zhang (2013) and McMahon et al. (2011) highlighted the potentially great impact from the philosophy of learning in Confucian Heritage and the ethic of filial piety that students are not encouraged for questioning and querying in front of others and tutors [85, 86]. In this regard, Loh and Teo (2017) showed unique learning characteristics of students in Hong Kong, China, Singapore, and several other Asian regions such as a teacher-centered approach with learning outlines, lesson plans, and contents mostly performed by tutors, minimal questions for keeping harmony in class, and implicit communication [87]. Zhao and McDougall (2008) found that Chinese students seemed to be more expressive and positive in using asynchronous E-learning when compared with traditional face-to-face learning [88]. Interestingly, the present study showed a higher preference for face-to-face learning when compared with ERL for Hong Kong students. Such discrepancies between the two studies can be partly explained by several aspects: 1) Online learning in Hong Kong, China, and the West can be different while the degree of language barrier of Chinese students studying in the West is supposed much higher than those Hong Kong students studying in local institutions with English instructions [88]; 2) The planned asynchronous online learning, blended learning and flip classroom are usually incorporated with both E-learning and face-to-face mode while all the face-to-face components were replaced by remote format using ERL and; 3) due to the cultural differences between Hong Kong and the mainland China, Hong Kong students showed the less stereotypical feature of typical Chinese learners [85]. Therefore, it is likely that Chinese

students, especially those studying in Hong Kong, are not reluctant to partially adopt E-learning for the regular planned course but ERL with 100% online learning is not as preferable as the traditional face-to-face format.

The second most impactful feature for student satisfaction was "Q19 instructor efforts to enhance students learning". During COVID-19 learners were required to conduct online learning from home and maintain adequate social distancing, limiting their opportunity for learner-learner interactions such as group discussions or collaborative tasks, especially during practical lessons. Meanwhile, subjects involving specialized skills (i.e., sports coaching, physical manual therapy techniques) heavily rely on non-verbal cues and kinesthetic feedback to consolidate theoretical knowledge and practical skills. Therefore, instructors play an important role in enhancing the concentration, motivation, and level of understanding of learners, and facilitating quality online learning, in particular when the class is conducted through video conferencing tools. This was in line with the study conducted by Fatani (2020) and Alqurashi (2018) which showed the importance of instructor presence and instructor-learner interaction based on the community of inquiry (COI) framework [20, 43].

When compared with ERL, students can provide mutual supports with each other through group class activities and self-initiated conversation if they have learning difficulties or confusion in the context of face-to-face learning. Without the immediate interaction and supports from the peers, feedback or emotional expressions from instructors, and interactive teaching styles through the use of video conferencing for ERL becomes extremely important to enhance cognitive presence including information processing, content digestion, and critical thinking [20, 43]. Moreover, as Carpenter et al. (2020) have pointed out the importance of engaging teaching with full enthusiasm for enhancing students' perceived learning while teacher-centered teaching seems to be more preferential for Chinese students, therefore, instructors need to be careful in judging the number of activities demanding students' interaction, and structured contents with fluent and engaging lectures [45]. Furthermore, ERL class setup such as the provision of additional resources for class preparation, proper camera and audio setting, the use of whiteboard for adding interactive learning on shared screen as well as the post-class support including the access of recorded ERL video may all be perceived as part of the instructors' tasks. In this regard, future studies on the influencing factors of perceived instructor efforts are warranted.

The third highest ranked feature in our study was "Q16 methods adopted for adjusting or modifying assessments". The recent thematic analysis conducted by Shim and Lee (2020) supported our findings that both inappropriate assessment proportion and ambiguity of assessment items were reported as areas of dissatisfaction [5]. Meanwhile, Hong Kong is known for its examination-oriented culture, and local Chinese students emphasize assessment results rather than genuine learning in general [89]. When ERL was enforced, all practical lessons (e.g., tutorials, workshops, laboratory sessions) were suspended or modified to accommodate the online format. Numerous on-site continuous assessments, project work, written and practical examinations were also changed to different formats, to support students in completing the academic year and graduating on time. Therefore, without any experience or mental preparation on the new assessment arrangement, it is expected that students had concerns about the validity, fairness, and relevancy of new assessment items.

Previous studies have inconsistent findings on the association between assessment materials and learner perceived satisfaction [8, 37]. Although based on the EESS model proposed by Al-Fraihat (2020), assessment materials are considered part of educational system quality which may contribute to the student satisfaction of E-learning, they have rejected their hypothesis in this regard while Cidral et al. (2018) supported the use of diversified assessment items for promoting better perceived satisfaction [8, 37]. Due to the limited and inconsistent research

findings as well as lack of a theoretical model to conclude in this area, it is speculated that both the sudden change of assessment format and unique learning culture in Hong Kong or Asian regions contribute to the high importance of appropriate assessment method in dictating the ERL success in our study. It is worth noting that when cutting all the mid-term or continuous assessments, students may feel excessively stressed and burdened on the single final assessment [5]. On the other hand, too many assessment items may also over-stress students in terms of quantity. Therefore, without lowering the level of assessments and learning quality, tutors and course creators may need to provide sufficient diversity and the optimal number of assessment items meanwhile considering the need for additional preparation time for students to complete all the required tasks during the crisis period.

The fourth most important feature was Q11 which concerning if the course or lesson activities were well delivered in an online environment. Students from different faculties and programs participated in lessons with varied styles and formats. For example, instructors might only provide online lectures for certain theory-oriented lessons, while interactive online learning could be used to replace practical or analytical based sessions. However, our study showed the lowest score in this item (3.91 out of 7, disagree to neutral) among all the features. When the technical issues are believed not to be the important factors determining student satisfaction while students are satisfied with instructors' effort in the present study, likely, the temporarily substituted class delivery and learning activities were not able to meet the intended learning outcomes and objectives. Since multiple departments were included and students came from the high diversity of courses such as engineering, design, sports, hotel, and culinary related programs in our study, practical and vocational oriented classes are supposed to be more demanding on face-to-face learning assisted with real-time feedback, collaborative works, on-site and hands-on practices. Therefore, it is unsurprised for the high predictive strength and low score for this question item.

The previous study conducted by Shim and Lee (2020) has provided some insights in this regard as their students reflected unilateral interactions (e.g. no interaction, no or difficulty with immediate feedback, difficult to ask questions or share thoughts), constraints on practice or experimental activities (e.g. reduced learning because of inadequate or no practice, the restricted or inconvenient arrangement in practical classes), constraints on team projects (e.g. inconvenient or lack of group activities) as well as a reduced understanding of classes as their major dissatisfaction items in ERL [5]. Although learner-learner interaction was not shown to be critical for dictating student satisfaction in online learning environments, both learner-instructor and learner-content interaction are important factors affecting the perceived satisfaction [43]. In their study, online course materials were deemed important to facilitate both the understanding of contents, the students' interest in the program, and relaying new knowledge to their own experience which all were in line with the cognitive presence under the COI framework [20]. However, practical learning and real-time feedback for further enhancement seem to be more important content than pre-recorded or disseminated course materials when the programs are practical and vocational oriented.

### 6.4 Theoretical and practical implications

With the accelerated use of ERL under the strong pressure induced by the COVID-19 pandemic, there is an urgent need to investigate the perception from students' perspectives in such a novel education strategy. This study aimed to firstly, find accurate models between machine learning algorithms and multiple linear regression, and subsequently predict student satisfaction on ERL with the selected model as well as identify relevant important features/predictors. It provided insights into both the statistical methods and relevant predictors in

determining the student satisfaction on ERL in higher education. The use of feature selection pre-processing techniques such as RF-RFE and the verification of model accuracy with testing dataset should be considered as necessary routine procedures to safeguard the selected model against inflated performance. With the more accurate predictive model and precise important features used for further analysis, the empirical results of this study provided insights to both teaching faculties, managerial staff, and administrative personnel for what should be focused on to maximize the learning experience and teaching performance using ERL during the crisis for higher education.

Through extensive literature review, although six constructs including readiness, accessibility, instructor-related, assessment-related, learning-related, and self-concerned, were proposed to be important for predicting student satisfaction on ERL, only very few items from instructor-related, assessment-related, and learning-related constructs were found to be critical. With the use of novel application of machine learning and feature selection method, RF-RFE, redundant and unimportant predictors were screened out and not included in the final production model. Previous studies such as the recent one conducted by Al-Fraihat et al. (2020) reported 71.4% of explained variance for their predictive model concerning their EESS model [8]. However, without dealing with multicollinearity issues and using the testing dataset for accuracy verification, the substantial explanatory power of their model could be questionable. In this regard, our ENet model has moderately explained 65.2% of the variance of student satisfaction.

Interestingly, our study did not fully support the multidimensional EESS model proposed by Al-Fraihat et al. (2020) [8]. However, our findings could be somewhat explained and supported by the interaction theory highlighted by Alqurshi (2018) especially the learner-instructor and learner-content interactions, and form cohesion with the COI framework addressed by Fatani (2020) [20, 43].

## 6.5 Limitations

Although this study attempted to predict the student satisfaction for ERL and identify important predictors, some limitations should be highlighted. One limitation is the limited question coverage in our survey. As the length of the questionnaire was controlled to avoid a low response rate due to the lengthy and boring survey process, questions from the original EDU-CAUSE survey kit were shortened, selected, and modified. Therefore, several items potentially affecting the students' satisfaction such as internship, grading structures, privacy issues, and missing out on on-campus activities were not included. Therefore, our ENet model could only explain 65.2% of variance concerning satisfaction score but the remained 35% of the variance of ERL perceived satisfaction was not investigated. Moreover, some questions in our study were broad and non-specific such as Q11 regarding if the course and learning activities were well delivered in an online environment. When it was found to be critical to predict student satisfaction, the potential underlying causes were not further investigated with followed up questions. Future research could base on the four important predictors detected in the present study to look into the specific causes, activities, or items affecting the satisfaction on online learning delivery, instructors' effort, modified assessment items, and reasons behind preferential learning with the face-to-face format for each faculty or program.

When using machine learning models for predictive purposes, it is worth noting that tuning of model parameters and nature of the dataset such as the number of predictors involved can affect the final results. Therefore, more similar studies with comparable or larger datasets are recommended to better confirm the similar findings. Since the current research was conducted at a single institution in Hong Kong addressing ERL, results cannot be generalizable to other

universities, countries, and other online approaches (i.e., blended or flipped learning). Furthermore, this paper only showed results from the students' perspective without including faculties and other supporting staff. Meanwhile, students from different faculties showed inconsistent preferences for face-to-face learning, therefore future studies may compare satisfaction scores and underlying predictors for learners and other school stakeholders from different programs, departments, or even institutions.

Last but not the least, our discussion only focused on the four most significant/important constructs shown in both multiple linear regression and ENet from machine learning whereas, several other predictors also contributed to the prediction of student satisfaction to a certain extent. The influence from other non-significant or relatively less important constructs should not be completely ignored.

## 7 Conclusion

The findings from this study indicate that higher education institutes should put a high emphasis on facilitating and improving the efforts made by instructors, modulation of assessment methods to better accommodate the workload, appropriateness, and fairness for all the sudden changes using ERL during a crisis, preparing contingency plan and other alternative learning activities or resources to supplement the inadequacy, or learning deficits in ERL. Moreover, it is important to rule out the underlying reasons why face-to-face learning is more preferable for students from the different program such that course providers and teachers can provide specific and tailor-made courses, learning activities, contents, and implementation methods to enhance the learning experience and maximize the students' satisfaction.

## Acknowledgments

We would like to thank Professor Christina Hong and Dr. Will Ma for initiating and leading this research project. There is no conflict of interest for authors or researchers involved in this project.

## Author Contributions

**Data curation:** Indy Man Kit Ho, Kai Yuen Cheong, Anthony Weldon.

**Formal analysis:** Indy Man Kit Ho, Kai Yuen Cheong.

**Investigation:** Indy Man Kit Ho, Kai Yuen Cheong, Anthony Weldon.

**Methodology:** Indy Man Kit Ho.

**Project administration:** Indy Man Kit Ho, Kai Yuen Cheong.

**Software:** Indy Man Kit Ho.

**Validation:** Indy Man Kit Ho, Kai Yuen Cheong.

**Visualization:** Indy Man Kit Ho.

**Writing – original draft:** Indy Man Kit Ho, Anthony Weldon.

**Writing – review & editing:** Indy Man Kit Ho, Kai Yuen Cheong, Anthony Weldon.

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
