## [Decision Letter · Decision Letter 0]

30 Nov 2020

PONE-D-20-32904

Predicting Student Satisfaction Scores after Emergency Remote Learning in Higher Education during COVID-19, with Selected Machine Learning and Statistical Methods

PLOS ONE

Dear Dr. Indy Man Kit Ho,

Thank you for submitting your manuscript to PLOS ONE. After careful consideration, we feel that it has merit but does not fully meet PLOS ONE’s publication criteria as it currently stands. Therefore, we invite you to submit a revised version of the manuscript that addresses the points raised during the review process.

We look forward to receiving your revised manuscript.

Kind regards,

Zaher Mundher Yaseen

Academic Editor

PLOS ONE

Journal Requirements:

2. Please update the Data Availability Statement to provide contact information for the Technological and Higher Education Institute of Hong Kong, where the data can be requested.

3.We suggest you thoroughly copyedit your manuscript for language usage, spelling, and grammar. If you do not know anyone who can help you do this, you may wish to consider employing a professional scientific editing service.  

4.Thank you for stating the following financial disclosure:

 [NO].

5.Thank you for stating the following in your Competing Interests section: 

[NO].

6.We note that you have indicated that data from this study are available upon request. PLOS only allows data to be available upon request if there are legal or ethical restrictions on sharing data publicly. For more information on unacceptable data access restrictions, please see http://journals.plos.org/plosone/s/data-availability#loc-unacceptable-data-access-restrictions.

7. Please ensure that you include a title page within your main document. You should list all authors and all affiliations as per our author instructions and clearly indicate the corresponding author.

Additional Editor Comments (if provided):

Author should pay attention very much for the reviewers comments. The manuscript is required as extensive revision.

Reviewers' comments:

Reviewer's Responses to Questions

**Comments to the Author**

1. Is the manuscript technically sound, and do the data support the conclusions?

Reviewer #1: Partly

Reviewer #2: Partly

2. Has the statistical analysis been performed appropriately and rigorously? 

Reviewer #1: Yes

Reviewer #2: Yes

3. Have the authors made all data underlying the findings in their manuscript fully available?

Reviewer #1: Yes

Reviewer #2: No

4. Is the manuscript presented in an intelligible fashion and written in standard English?

Reviewer #1: No

Reviewer #2: No

5. Review Comments to the Author

Reviewer #1: * The title is very long and not informative.

* The abstract is misleading and does not convey the main purpose of the study.

* The logical stance of the research problem and its corresponding research objective is not clearly designated in the introduction section.

* Some grammatical mistakes and language logical flow were spotted in the manuscript, and therefore, it should be sent out for proofreading.

* Sections 2 to 4 can be merged under one main section (e.g., background or literature review).

* Collecting data through surveys are sometimes exposed to bias. How did you handle this issue? This needs to be clarified briefly in the manuscript.

* The discussion section has mainly tackled the statistical analysis of the findings, which sounds good. However, I would expect to discuss the results of your research and link them to the prior research. Why they are important?

* What are the theoretical contributions and practical implications of your research? This should be clearly specified in a separate section before the conclusion.

* The figures are not clear. Please provide high-resolution figures.

* The following relevant references were missed in your manuscript. You can refer to them and include them in your literature review section:

- Saa, A. A., Al-Emran, M., & Shaalan, K. (2019). Factors affecting students’ performance in higher education: a systematic review of predictive data mining techniques. Technology, Knowledge and Learning, 24(4), 567-598.

- Saa, A. A., Al-Emran, M., & Shaalan, K. (2019, March). Mining student information system records to predict students’ academic performance. In International conference on advanced machine learning technologies and applications (pp. 229-239). Springer, Cham.

Reviewer #2: Reviewer Comments to Author:

The study is quite interesting, and the outcomes of this paper are a valuable addition to literature (after major changes/revision). This study investigates the feature selection technique of random forest feature recursive elimination and checking overfitting issues with training and testing data sets and identified improved accuracy for all selected models. The author allegedly argues that The most important predictors to determine student satisfaction scores during emergency remote learning includes: 1) preference of face-to-face learning; 2) instructors' efforts to enhance students learning; 3) adjusted assessment was appropriate, and 4) online course activities were well delivered. This paper contributes to the literature by showing evidence of distance learning; social distancing practices are limited to different countries' data (The author should disclose country names). Overall, this article is not written clearly, but it should not be granted a publication in PLOS ONE. There are problems with English, i.e., Punctuations, grammar, and propositions. The paper needs copy editing. The introduction section, critical literature, and discussion of findings are major and significant concerns of this article. See comments below for details.

The below are major concerns and areas of improvement with suggestions.

1. Originality: The paper aims to investigate the feature selection technique of random forest feature recursive elimination, checking overfitting issues with training and testing data sets, and identifying improved accuracy for all selected models. The document did not contain significant new information appropriate to justify the publication. There is a need to revise the paper title, and the current title does not portray the paper's actual meaning.

2. Abstract: The abstract is not well written. There is a need to revise the abstract with explicit contents, i.e., the main issue, sampling, a statistical tool, methods, results, and implication. The author’s should provide a precisely and focused abstract.

• What is the practical and theoretical contribution of this article to literature?

• The sampling criteria, population, and unit of analysis for the selection of students are missing. The author should highlight the sampling criteria for more clarity to readers.

• As a suggestion for improvement, the author should not use the same Keywords as Paper Title. It is encouraged to used different keywords that are not in the Paper title. It will enhance paper searchability after the publication.

3. Introduction: The introduction section is not well written. There are ambiguous statements and no clarity in the introduction section.

• The introduction section is not started with a broader area and issue or in a global context. There is no synthesis in writing an introduction section.

• There is less debate on the targeted country problem, i.e., countries problems. It will add more value to the paper if the author explains some statistics and recent issues.

• In the Introduction section, the brief discussions of methods, tools, sampling, and findings are missing.

• An important question to answer is, “Why should PLOS ONE readers be interested in the results of this paper, which scrutinized only some countries' data?” The reason given is not supportive. Are the findings generalizable to other developing countries like India and Indonesia? The author(s) needs to improve underwritten motivation.

• Author(s) did not follow the academic paper's standard content, i.e., Introduction, Literature Review; follow as per journal requirements and guidelines.

• There is no roadmap at the end of the introduction that conveys the rest of the paper's structure.

4. Relationship to Literature: The paper incorporated major literature of learning culture, but the paper does not sufficiently cover recent research in the area. Helpful in this regard would be to include relevant research recently covered in top journals of similar scope. Further, work needs to be done to support the findings based on the current literature, as a recent theory in the area is directly counter to what was found.

• Author(s) did not use any underpinning theory to justify this research, i.e., resource dependency, agency theory etc. This is the major concern of this paper; the author(s) should highlight how this research contributes to the theory or the contribution of the theory. The authors did not discuss much with the use of developed theories in this area.

• There is a need to add more critical recent literature and based on theoretical argumentation.

5. Methodology: The author did not develop its argument from the appropriate theory and explored models previously studied in the same area. However, the data is focused on students. The article represents primary data from the survey technique as the sample and methodology and relevant to present the proposed theme's context. However, the empirical sample was fair enough to illustrate significant empirical results of this paper, and the methods employed are appropriate. In addition, there are major concerns with sampling design and data nature.

(i) What is the population of students (Sampling frame, total population, etc.)?

(ii) What are the criteria for the selection of firms for data collection? (sampling techniques, i.e., random. Cluster or judgmental sampling)?

(iii) Regarding the methodology, more details and justification of why Machine Learning analysis is used?

(iv) Author (s) did not define the data collection and sampling in a clear way.

6. Results: The analysis is provided to tie up with the findings.

• The author should provide complete statistical analysis, i.e., descriptive analysis, including skewness and kurtosis.

7. Discussion and findings: As results are clearly provided. However, there is a substantial discussion on results.

• Author(s) should discuss the limitations of this study and future research direction in a constructive way. Hence, authors should write in prices and in a constructive way under a subsection of discussion.

• Author(s) did not discuss the theoretical and practical contribution of this study. The author (s) should discuss this study's theoretical and practical contribution in the separate subsection under discussion for more clarity.

8- Conclusion: The author should provide concluding statements rather than repetitive statements in the conclusion portion.

9. Citation and End References

• The in-text citations and end list of references do not sufficiently correspond.

• The author did cite the latest literature relevant to the target issue in this paper. The reviewer found that the author has cited only eight (8) recently published papers in this article (Most of the cited articles ten years old). As a suggestion, the author must cite new articles (latest literature) to make holistic discussion and sturdy paper with high readability

11. Quality of Communication: The paper needs further proofreading. I have tried to read the paper constructively, but I felt it suffers from poor writing. I, therefore, request the author to pass the manuscript for professional proofreading. I suggest that a more careful investigation of prior literature can make this paper distinguishable. Linking this article with prior studies does not seem sufficient, which weakens the justification of incremental contributions.

Suggested Revisions

• The manuscript needs a professional proofreading.

• Abstract revision (complete / Major concern)

• Sampling and Data (Need to explain in detail / Major concern)

• Introduction Section (Please re-write holistically)

• There is a need to discuss the theoretical and practical contributions of this study in separate subsections.

• Conclusion section need to add and revise

I hope that the comments provided can help in this regard.

6. PLOS authors have the option to publish the peer review history of their article (what does this mean?). If published, this will include your full peer review and any attached files.

Reviewer #1: No

Reviewer #2: No

---

## [Author Response · Author response to Decision Letter 0]

17 Feb 2021

We thank the two reviewers for their time and valuable suggestions. In this revised version we have answered all the questions raised by the reviewers and edited the manuscript with substantial revision on English syntax accordingly. We hope that this revised manuscript meets the standard for publication in PLOS ONE. Below please find our point-to-point responses to reviewers.

Reviewer #1: * The title is very long and not informative.

* The abstract is misleading and does not convey the main purpose of the study.

Response: Thanks for the comment. The abstract was rewritten.

* The logical stance of the research problem and its corresponding research objective is not clearly designated in the introduction section.

Response: Thanks much for this. The introduction section (line 72-127) was substantially revised while an extensive literature review (Line 129-359) as section 2 was added. By reviewing the relevant theories from previous studies, the three research questions were formulated to show more logical and clearer research objectives (Line 361-377)

* Some grammatical mistakes and language logical flow were spotted in the manuscript, and therefore, it should be sent out for proofreading.

Response: The manuscript was proof read by a native English speaker and hope the revised edition is more up to standard.

* Sections 2 to 4 can be merged under one main section (e.g., background or literature review).

Response: Thanks for the comment. As mentioned before, now only an extensive literature review (line 129-359) with some of the original contents from section 2 to 4 merged.

* Collecting data through surveys are sometimes exposed to bias. How did you handle this issue? This needs to be clarified briefly in the manuscript.

Response: Thanks much for the comment. We have added the relevant contents from line 386-396

* The discussion section has mainly tackled the statistical analysis of the findings, which sounds good. However, I would expect to discuss the results of your research and link them to the prior research. Why they are important?

Response: Thanks much for the comment. We have added the relevant contents from line 609-752

* What are the theoretical contributions and practical implications of your research? This should be clearly specified in a separate section before the conclusion.

Response: Thanks much for the comment. We have added the relevant contents from line 725-753

* The figures are not clear. Please provide high-resolution figures.

Response: Thanks for reminder. The figures for replaced by 300 dpi now.

* The following relevant references were missed in your manuscript. You can refer to them and include them in your literature review section:

- Saa, A. A., Al-Emran, M., & Shaalan, K. (2019). Factors affecting students’ performance in higher education: a systematic review of predictive data mining techniques. Technology, Knowledge and Learning, 24(4), 567-598.

- Saa, A. A., Al-Emran, M., & Shaalan, K. (2019, March). Mining student information system records to predict students’ academic performance. In International conference on advanced machine learning technologies and applications (pp. 229-239). Springer, Cham.

Response: Thanks for the information provided. The former one for the systematic review was added while the second one (conference paper) was not used. Meanwhile, per the request from another reviewers, I have added more than 10 very recent literature to support the theory or idea.

Reviewer #2: Reviewer Comments to Author:

The study is quite interesting, and the outcomes of this paper are a valuable addition to literature (after major changes/revision). This study investigates the feature selection technique of random forest feature recursive elimination and checking overfitting issues with training and testing data sets and identified improved accuracy for all selected models. The author allegedly argues that The most important predictors to determine student satisfaction scores during emergency remote learning includes: 1) preference of face-to-face learning; 2) instructors' efforts to enhance students learning; 3) adjusted assessment was appropriate, and 4) online course activities were well delivered. This paper contributes to the literature by showing evidence of distance learning; social distancing practices are limited to different countries' data (The author should disclose country names). Overall, this article is not written clearly, but it should not be granted a publication in PLOS ONE. There are problems with English, i.e., Punctuations, grammar, and propositions. The paper needs copy editing. The introduction section, critical literature, and discussion of findings are major and significant concerns of this article. See comments below for details.

The below are major concerns and areas of improvement with suggestions.

1. Originality: The paper aims to investigate the feature selection technique of random forest feature recursive elimination, checking overfitting issues with training and testing data sets, and identifying improved accuracy for all selected models. The document did not contain significant new information appropriate to justify the publication. There is a need to revise the paper title, and the current title does not portray the paper's actual meaning.

Response: Thanks much for the comments. The title was revised while the values/insights of this paper were also highlighted in both introduction (Line 72-127) and discussion (Line 546-752)

2. Abstract: The abstract is not well written. There is a need to revise the abstract with explicit contents, i.e., the main issue, sampling, a statistical tool, methods, results, and implication. The author’s should provide a precisely and focused abstract.

Response: Thanks much for the comments. The abstract (Line 38-53) was rewritten to match the primary and secondary objectives of the study as well as the title.

• What is the practical and theoretical contribution of this article to literature?

Response: Thanks much for pointing out this. A separate sub-section regarding this was added in discussion (725-752) per the request from another reviewer as well.

• The sampling criteria, population, and unit of analysis for the selection of students are missing. The author should highlight the sampling criteria for more clarity to readers.

Response: Thanks for the comment. The method section regarding this part was revised to address this (Line 389-408).

• As a suggestion for improvement, the author should not use the same Keywords as Paper Title. It is encouraged to used different keywords that are not in the Paper title. It will enhance paper searchability after the publication.

Response: Thanks for the reminder. Some keywords were changed

3. Introduction: The introduction section is not well written. There are ambiguous statements and no clarity in the introduction section.

• The introduction section is not started with a broader area and issue or in a global context. There is no synthesis in writing an introduction section.

• There is less debate on the targeted country problem, i.e., countries problems. It will add more value to the paper if the author explains some statistics and recent issues.

• In the Introduction section, the brief discussions of methods, tools, sampling, and findings are missing.

• An important question to answer is, “Why should PLOS ONE readers be interested in the results of this paper, which scrutinized only some countries' data?” The reason given is not supportive. Are the findings generalizable to other developing countries like India and Indonesia? The author(s) needs to improve underwritten motivation.

• Author(s) did not follow the academic paper's standard content, i.e., Introduction, Literature Review; follow as per journal requirements and guidelines.

• There is no roadmap at the end of the introduction that conveys the rest of the paper's structure.

Response: Thanks much for the pointing this out. The introduction was heavily rewritten such that only a broad/global context was given as the background of this study. For specific contents, subsections were made throughout the extensive literature review in section 2. A roadmap to express the structure of the paper was given at the end of the introduction.

However, we found that it is very challenging to collect data in multiple countries/regions regarding Emergency Remote Learning during pandemic given the short timeframe. Meanwhile, similar studies focusing on single country/institution were found in paper such as Shim and Lee (2020) focused on Korean students; Fatani (2020) studied 162 students in Saudi; Almusharraf & Khahro (2020) for Saudi; Shahzad et al. (2020) for Malaysian students. Since none of the previous study regarding ERL during pandemic focus on Chinese, our paper can give added value while in the long run, future research using systematic review may make use of all these data from different region to give more conclusive insights and knowledge. Therefore, we understand that our data/findings are not without limitations and these were highlighted in the limitation section as well (Line 754-783). In addition, the potential unique learning culture for Chinese/Hong Kong students were highlighted in the discussion section to give more insight to explain the observed findings. (Line 630-647)

4. Relationship to Literature: The paper incorporated major literature of learning culture, but the paper does not sufficiently cover recent research in the area. Helpful in this regard would be to include relevant research recently covered in top journals of similar scope. Further, work needs to be done to support the findings based on the current literature, as a recent theory in the area is directly counter to what was found.

• Author(s) did not use any underpinning theory to justify this research, i.e., resource dependency, agency theory etc. This is the major concern of this paper; the author(s) should highlight how this research contributes to the theory or the contribution of the theory. The authors did not discuss much with the use of developed theories in this area.

• There is a need to add more critical recent literature and based on theoretical argumentation.

Response: Thanks much for raising this. An extensive literature review as section 2 was given (Line 129-359). We have added 40+ more literature to consolidate the theory base to support the research questions and research objectives while quite a number of those newly added literature were quoted from very recent and high quality journals

5. Methodology: The author did not develop its argument from the appropriate theory and explored models previously studied in the same area. However, the data is focused on students. The article represents primary data from the survey technique as the sample and methodology and relevant to present the proposed theme's context. However, the empirical sample was fair enough to illustrate significant empirical results of this paper, and the methods employed are appropriate. In addition, there are major concerns with sampling design and data nature.

(i) What is the population of students (Sampling frame, total population, etc.)?

(ii) What are the criteria for the selection of firms for data collection? (sampling techniques, i.e., random. Cluster or judgmental sampling)?

(iii) Regarding the methodology, more details and justification of why Machine Learning analysis is used?

(iv) Author (s) did not define the data collection and sampling in a clear way.

Response: Thanks much for this. The method section about this was revised (Line 379-408) meanwhile, the introduction (Line 112-121) and discussion (Line 548-607) and (Line 727-737) can supplement the reason for using machine learning.

6. Results: The analysis is provided to tie up with the findings.

• The author should provide complete statistical analysis, i.e., descriptive analysis, including skewness and kurtosis.

Response: Thanks much for this. Table 5 regarding internal reliability (Line 486-487) was added while skewness and kurtosis were given in table 4 (Line 485-486)

7. Discussion and findings: As results are clearly provided. However, there is a substantial discussion on results.

• Author(s) should discuss the limitations of this study and future research direction in a constructive way. Hence, authors should write in prices and in a constructive way under a subsection of discussion.

• Author(s) did not discuss the theoretical and practical contribution of this study. The author (s) should discuss this study's theoretical and practical contribution in the separate subsection under discussion for more clarity.

Response: Thanks much for this. Limitations and implications were added as subsections of discussion section (Line 725-783)

8- Conclusion: The author should provide concluding statements rather than repetitive statements in the conclusion portion.

Response: Thanks much for the comment on this. The conclusion section was rewritten (Line 785-793)

9. Citation and End References

• The in-text citations and end list of references do not sufficiently correspond.

• The author did cite the latest literature relevant to the target issue in this paper. The reviewer found that the author has cited only eight (8) recently published papers in this article (Most of the cited articles ten years old). As a suggestion, the author must cite new articles (latest literature) to make holistic discussion and sturdy paper with high readability

Response: Thanks much for the comment on this. As mentioned in previous part, 40+ extra reference with quite a number from high quality journal and very recent (2017-2020) were added

11. Quality of Communication: The paper needs further proofreading. I have tried to read the paper constructively, but I felt it suffers from poor writing. I, therefore, request the author to pass the manuscript for professional proofreading. I suggest that a more careful investigation of prior literature can make this paper distinguishable. Linking this article with prior studies does not seem sufficient, which weakens the justification of incremental contributions.

Response: Thanks much for the comment on this. The manuscript was proofread by a native English speaker. However, if it is still not up to publishable standard, I am willing to further send out to seek for more English editing and revision.

Suggested Revisions

• The manuscript needs a professional proofreading.

• Abstract revision (complete / Major concern)

• Sampling and Data (Need to explain in detail / Major concern)

• Introduction Section (Please re-write holistically)

• There is a need to discuss the theoretical and practical contributions of this study in separate subsections.

• Conclusion section need to add and revise

I hope that the comments provided can help in this regard.

---

## [Decision Letter · Decision Letter 1]

18 Mar 2021

Predicting Student Satisfaction of Emergency Remote Learning in Higher Education during COVID-19 using Machine Learning Techniques

PONE-D-20-32904R1

Dear Dr. Ho,

We’re pleased to inform you that your manuscript has been judged scientifically suitable for publication and will be formally accepted for publication once it meets all outstanding technical requirements.

Kind regards,

Zaher Mundher Yaseen

Academic Editor

PLOS ONE

Additional Editor Comments (optional):

Reviewers' comments:

Reviewer's Responses to Questions

**Comments to the Author**

1. If the authors have adequately addressed your comments raised in a previous round of review and you feel that this manuscript is now acceptable for publication, you may indicate that here to bypass the “Comments to the Author” section, enter your conflict of interest statement in the “Confidential to Editor” section, and submit your "Accept" recommendation.

Reviewer #1: All comments have been addressed

2. Is the manuscript technically sound, and do the data support the conclusions?

Reviewer #1: Yes

3. Has the statistical analysis been performed appropriately and rigorously? 

Reviewer #1: Yes

4. Have the authors made all data underlying the findings in their manuscript fully available?

Reviewer #1: Yes

5. Is the manuscript presented in an intelligible fashion and written in standard English?

Reviewer #1: Yes

6. Review Comments to the Author

Reviewer #1: The authors have addressed all the requested comments. Therefore, I recommend the manuscript for publication.

7. PLOS authors have the option to publish the peer review history of their article (what does this mean?). If published, this will include your full peer review and any attached files.

Reviewer #1: No

---

## [Editor Report · Acceptance letter]

23 Mar 2021

PONE-D-20-32904R1 

Predicting Student Satisfaction of Emergency Remote Learning in Higher Education during COVID-19 using Machine Learning Techniques 

Dear Dr. HO:

I'm pleased to inform you that your manuscript has been deemed suitable for publication in PLOS ONE. Congratulations! Your manuscript is now with our production department. 

Kind regards, 

on behalf of

Dr. Zaher Mundher Yaseen 

Academic Editor

PLOS ONE